# “Small Hepatocytes” in the Liver

**DOI:** 10.3390/cells12232718

**Published:** 2023-11-27

**Authors:** Toshihiro Mitaka, Norihisa Ichinohe, Naoki Tanimizu

**Affiliations:** 1Department of Tissue Development and Regeneration, Institute of Regenerative Medicine, Sapporo Medical University School of Medicine, Sapporo 060-8556, Japan; nichi@sapmed.ac.jp (N.I.); tanimizu@g.ecc.u-tokyo.ac.jp (N.T.); 2Division of Regenerative Medicine, Center for Stem Cell Biology and Regenerative Medicine, The Institute of Medical Science, The University of Tokyo, Tokyo 108-8639, Japan

**Keywords:** liver stem/progenitor cells, regeneration, cell transplantation, proliferation, differentiation, self-renewal

## Abstract

Mature hepatocytes (MHs) in an adult rodent liver are categorized into the following three subpopulations based on their proliferative capability: type I cells (MH-I), which are committed progenitor cells that possess a high growth capability and basal hepatocytic functions; type II cells (MH-II), which possess a limited proliferative capability; and type III cells (MH-III), which lose the ability to divide (replicative senescence) and reach the final differentiated state. These subpopulations may explain the liver’s development and growth after birth. Generally, small-sized hepatocytes emerge in mammal livers. The cells are characterized by being morphologically identical to hepatocytes except for their size, which is substantially smaller than that of ordinary MHs. We initially discovered small hepatocytes (SHs) in the primary culture of rat hepatocytes. We believe that SHs are derived from MH-I and play a role as hepatocytic progenitors to supply MHs. The population of MH-I (SHs) is distributed in the whole lobules, a part of which possesses a self-renewal capability, and decreases with age. Conversely, injured livers of experimental models and clinical cases showed the emergence of SHs. Studies demonstrate the involvement of SHs in liver regeneration. SHs that appeared in the injured livers are not a pure population but a mixture of two distinct origins, MH-derived and hepatic-stem-cell-derived cells. The predominant cell-derived SHs depend on the proliferative capability of the remaining MHs after the injury. This review will focus on the SHs that appeared in the liver and discuss the significance of SHs in liver regeneration.

## 1. Introduction

The organs that make up an individual are known to have their own specific homeostasis and the turnover rates of their constituent cells differ from each other. Cell renewal has been extensively studied in tissues with a fast turnover, such as the epidermis, blood, and gastrointestinal tract [1,2,3]. A stem-cell-dependent cell supply system has been established in tissues with a fast turnover, such as epithelial cells of the small intestine and skin, where cells are replaced in approximately 1–4 weeks. In contrast, liver homeostasis is generally not maintained by the cell supply from the stem cell system.

Mature hepatocytes (MHs) have a long life span of approximately 6 months in rodents and 2 years in humans [4,5,6]. The liver has a variety of physiological functions and a strong regenerative capacity. The remaining liver temporally reserves the capacity to maintain homeostasis of the organism and eventually recovers its original mass by the division of remaining hepatocytes, which takes 7–10 days in rodents, even if 2/3 of its functions are lost due to surgical or drug-induced injury.

The stem/progenitor cell system residing in the liver is generally activated when liver function is severely impaired and cannot be immediately restored. Oval cells (OCs) and small hepatocytes (SHs) are known as liver stem and progenitor cells, respectively [7,8,9,10]. OCs are distinguished by their oval nuclei and scanty cytoplasm [11]. Small epithelial cells radiate from the portal region, forming tubular structures along the hepatocytic cord or small clusters outside the portal region. OCs express markers of biliary epithelial cells (BECs) [keratin (Krt) 7/19, OV-6, OC-2, EpCAM, Sox9, Connexin [Cx] 43], hepatoblasts [α-fetoprotein (AFP), γ-glutamyltranspeptidase, protein delta homolog 1/preadipocyte factor 1 (Dlk)1, muscle pyruvate kinase], and hematopoietic stem cells (c-Kit, CD34, flt-3, Thy1) [8,12,13,14,15,16,17]. The cells are believed to be derived from the Canal of Hering (CoH), where small bile ductules are connected to bile canaliculi (BCs) that are formed between hepatocytes, or from mesenchymal cells around small bile ductules in the Glisson’s sheath. OCs can differentiate into MHs via SHs. OC-derived SHs appear in some of the small bile ducts or the periportal area adjacent to the small bile ducts. Except for their small size, these cells are polygonal in shape with round nuclei and prominent nucleoli and are morphologically indistinguishable from typical hepatocytes. Additionally, the cells express both α-fetoprotein (AFP) and albumin [8,15,18]. However, OCs are not always activated when liver regeneration is severely impaired due to hepatocyte loss or injury. OC activation depends on whether the remaining hepatocytes can proliferate and recover liver function immediately after injury. SHs appear transiently and deeply involved in liver regeneration, regardless of whether OCs are activated or if the remaining hepatocytes proliferate on their own.

SHs are cells that are morphologically identical to hepatocytes except for their size, but much smaller than typical MHs. Both in vitro [19,20] and in vivo [8,21,22] studies have demonstrated the existence of SHs. Many excellent reviews on OCs have been reported [12,13,23,24,25,26,27,28,29,30,31,32,33], whereas this review will focus primarily on the “small hepatocytes (SHs)” that appear in the rat livers and discuss how, when, and why these cells appear in the liver, as well as the role of SHs in liver diseases.

## 2. Growth Capability of Hepatocytes

Postnatal liver growth is characterized by remarkably slow hepatocyte proliferation. The turnover rate of normal adult human hepatocytes has been estimated to be 1 in 20,000–40,000 cells at any given time [4]. Therefore, the life span of the cells has been estimated to be 200–400 days and to be replaced approximately once a year [4,5,6]. However, the mitotic activity of hepatocytes is very high in the suckling period of rodents, in which almost all the hepatocytes actively proliferate and the liver mass rapidly expands [34]. Thereafter, hepatocytes exhibit heterogeneity in their proliferative potential, although the cells with almost the same proliferative capability as hepatoblasts are distributed in the whole lobules. At birth, 16–18% of hepatocytes synthesize DNA with a uniform distribution throughout the liver lobules [35,36]. DNA synthesis of rat hepatocytes rapidly decreases in hepatic zone 3 as early as day 1, followed by a decrease in zone 2 after 3 days. Only 6% of hepatocytes synthesize DNA by the age of 10–12 days, and 80% of these cells are localized in zone 1. Less than 0.2% of the hepatocytes synthesize DNA by 24 weeks of age [36].

Generally, the in vitro proliferative capability of hepatocytes progressively declines with maturation. A large proportion of suckling hepatocytes undergo DNA synthesis and mitosis under optimal hormonal conditions, whereas adult hepatocytes exhibit only limited DNA synthesis or mitosis [37]. Additionally, we revealed that almost all hepatocytes isolated from 3-week-old rat livers could equally proliferate at least until day 6, whereas 4-week-old hepatocytes began to show heterogeneity in their proliferative capability [38]. The high pulse labeling rate of 5–25% at birth declined steadily to approximately 0.1% in the adult rat liver because the number of proliferating cells gradually decreased with age [39,40]. Furthermore, the number of mitoses and the ^3^H-thymidine incorporation into DNA were much lower in aged rat hepatocytes than in the young ones [41,42,43,44]. These results indicate that a population of hepatocytes that cease to divide increases with age, which means that the limited number of proliferating hepatocytes go through more cycles to accomplish regeneration [45]. In particular, the heterogeneity of growth capability is manifested in MHs after weaning.

SHs have been identified as proliferating cells with hepatic characteristics. We initially found a remarkable increase in mononucleate cells within the culture of primary rat hepatocytes in the presence of 10 mM nicotinamide and epidermal growth factor (EGF) [19,46]. Thereafter, we observed SHs in primary cultured human hepatocytes isolated from surgically removed liver tissues (Figure 1Ad) [47]. One SH can clonally proliferate to form a colony while maintaining the characteristics of hepatocytes and the capability to differentiate into MHs (Figure 1Aa) [20,48,49]. SHs can be isolated from a healthy adult rat [20,48,50,51,52], mouse [53,54,55], porcine [56], and human livers (Figure 1Af) [47,57]. The population of SHs in an adult rat liver is approximately 1.5–2.0% of hepatocytes, and the number of the cells decreases with age [38,58]. SHs can rapidly proliferate to form a colony, and the average number of cells in a colony reaches approximately 27 cells at 10 days [48]. Additionally, Tateno and Yoshizato [58] reported that a single SH could generate up to 1400 cells, which indicates approximately 10 doublings of SHs. Furthermore, we confirmed that the SH subpopulation with a high growth potential could divide over 50 times in 17 weeks every 4-week passage [59]. Therefore, we regard the SH as a committed hepatocyte progenitor.

While in vitro experiments demonstrated the existence of MHs with a high growth potential, evidence is accumulating that hepatocytes possess tremendous replication potential in vivo using the mice bearing a transgene of the albumin-urokinase-type plasminogen activator (Alb-uPA) coding sequence fused to the albumin enhancer/promoter. Approximately half of the newborns died within several days after birth by fatal hemorrhaging, but the remaining transgenic offspring survived [60]. Mouse livers that survived demonstrated that transgene-deficient hepatocytes selectively proliferated to form clonal hepatic nodules, which finally replaced the entire liver [61]. As few as two nodules could reconstitute over 90% of liver mass in some cases. Furthermore, the nodules of both donor-derived hepatocytes and transgene-deficient endogenous ones clonally expanded to occupy a large area of the liver at the expense of the damaged hepatocytes when hepatocytes carrying the *β-galactosidase (β-gal)* gene were transplanted into Alb-uPA mouse livers [62]. The authors estimated that <7% of donor cells could give rise to nodules, indicating that the donated and endogenous hepatocytes underwent extensive growth, dividing at least 12 times. Additionally, transplantation of mouse hepatocytes into mice livers that carry a major urinary protein-urokinase-type plasminogen activator (MUP-uPA) fusion transgene demonstrated that donor cells had undergone approximately 12 cell-doubling events after transplantation [7]. Accumulation of the toxic intermediate fumarylacetoacetate damages hepatocytes in mice deficient of the tyrosine carbolic enzyme, fumarylacetoacetate hydrolase (FAH), and the mice die soon after birth [63,64]. As few as 1000 hepatocytes were reported to be sufficient to rescue the FAH-deficient mice and restore liver function when they were transplanted into FAH-deficient livers [65]. Additionally, studies using serial transplantation of mouse hepatocytes revealed that donated hepatocytes could divide >69 times [66]. The upper limit of the proliferative capacity of MHs remains unknown, but a subpopulation of MHs, probably SHs, certainly possesses a tremendous capacity for cell divisions. Transplantation of a relatively small number of MHs that possess a high growth capability may rescue patients suffering from severe liver diseases when the methods to improve the efficiency of engraftment are established, considering that 28 cell divisions are sufficient to reconstitute the entire liver from a single hepatocyte.

## 3. Subpopulation of MHs and Their Distribution in the Liver Lobules

We previously proposed at least three subpopulations of hepatocytes in an adult rat liver [9]: type I cells (MH-I), which are committed progenitor cells that possess a high growth capability and basal hepatocytic functions that can further differentiate into MHs; type II cells (MH-II), of which possible cell divisions is limited; type III cells (MH-III), which lose the ability to divide (replicative senescence) and reach the final differentiated state.

Almost all hepatocytes (MH-I), which have a proliferative capacity similar to hepatoblasts, divide and proliferate actively, and the liver grows rapidly immediately after birth (Figure 2). Heterogeneity in the proliferative potential of hepatocytes emerges during weaning. After weaning, MH-I proliferates rapidly, while a part of the cells becomes limited in its proliferative capacity. Those hepatocytes become MH-II, resulting in a rapidly diminishing population of MH-I. Some of the MH-I retain their progenitor cell capacity and become randomly distributed within whole liver lobules. MH-II reaches the replicative senescence with aging, thereby increasing the MH-III population. Generally, aged rat MHs have decreased proliferative capacity due to increased *Activin A* (*ActA)* and *p15INK4b* expressions compared to young ones [67]. Additionally, small foci of SHs are often observed in aged rat livers (Figure 1Cc) [68]. The foci contain small-sized hepatocytes and are distributed throughout the lobule. This finding indicates that MH-I may be selectively proliferating in the liver of aging rats because small numbers of MH-II are unable to compensate for all the lost cells.

Studies have revealed the random distribution of actively proliferating hepatocytes within the lobules, using mosaic livers from *spf^ash^* heterozygous female mice [69]. The *spf^ash^* mutation is located on the X chromosome and causes neonatal hyperammonemia in homozygous females due to ornithine transcarbamylase (OTC) deficiency [70]. Mosaics of wild-type hepatocytes and mutated hepatocytes expressing one-tenth of the OTC protein are found in heterozygous livers. During development, the hepatocytes proliferated and migrated extensively, but the progeny showing the same OTC expression intensity as the parent cells in the adult liver were present in contact with the parent ones, and the direction of their division was random, thereby forming a large, three-dimensionally contiguous cluster of hepatocytes [71,72].

Conversely, Kennedy S et al. [73] used a transgenic mouse, in which *β-gal* was expressed downstream of the human α1-antitrypsin (AT) promoter, to study the proliferative potential of MHs. Singles or doublets of *β-gal*-positive hepatocytes were randomly scattered throughout the liver in neonatal mice. The *β-gal*-positive cells formed larger clusters with aging and showed no predilection for the periportal or pericentral regions. Furthermore, Bralet MP et al. [74] labeled hepatocytes by injecting a retroviral vector carrying the *β-gal* gene into adult rat livers after 2/3 partial hepatectomy (PH). The labeled cells formed clusters with age, but their distribution within the lobule did not change throughout life, although several labeled cells tended to be located near periportal regions (zones 1 and 2). These results support our hypothesis of random distributions of MH-I throughout the lobule and oppose the “hepatocytes streaming theory” [75,76,77,78], which speculates that periportal stem cells give rise to daughter cells that stream toward the pericentral region and undergo apoptosis.

## 4. SHs In Vitro

### 4.1. Culture Medium

Rat SHs in a healthy adult liver are enriched by centrifugation at 150× *g* after eliminating MHs by repeated centrifugation at 50× *g*. The colonies appear at 4–5 days after plating, and SHs continue to expand (Figure 1Aa) [19]. The colonies that consist of SHs were also observed in the primary culture of rat hepatocytes in the serum-free chemically defined medium [79]. The authors demonstrated that the medium that contains diferric transferrin and arginine, as well as nicotinamide, was required for the sustained clonal proliferation of rat hepatocytes, resulting in SH expansion. Hepatocyte growth factor (HGF), transforming growth factor (TGF)-α, and/or EGF were also necessary to stimulate clonal growth. Additionally, the supplementation of nicotinamide in the medium could allow mouse primary hepatocytes to proliferate and survive for >2 months while maintaining their differentiated functions [55]. The proliferating hepatocytes with near-diploid features were immortalized after the long-term cultivation.

The emergence of SHs in the primary culture of hepatocytes was observed in the serum-free amino-acid-rich (Leibovitz 15, L-15) medium supplemented with NaHCO_3_ and EGF in a 5% CO_2_/95% air incubator, in addition to the condition with high nicotinamide concentrations [80]. SH colonies were often observed in the culture and the SHs continued to proliferate 4–5 days after plating. L-15 is a medium that was originally formulated to make a medium designed for use in a non-bicarbonate buffer system. The buffer with high concentrations of amino acids, especially arginine, was developed to culture cells without the use of a CO_2_ incubator [81]. Therefore, we added 20 mM NaHCO_3_ in the medium to culture hepatocytes in a 5% CO_2_/95% air incubator [80]. The L-15 medium contains approximately three times as many amino acids as Dulbecco’s modified Eagle medium (DMEM). High amino acid and insulin concentrations in hepatocyte cultures are required for protein synthesis and proteolysis inhibition [82]. Additionally, exogenous arginine may be essential for maintaining the proliferative capacity of hepatocytes in culture because proliferating hepatocytes lose their hepatic differentiation function, including arginine production through the urea cycle [79].

The importance of trace elements is well-known in a serum-free hepatocyte culture [9]. The original L-15 medium does not contain trace elements such as CuSO_4_, FeSO_4_, MnSO_4_, ZnSO_4_, and Na_2_SeO_3_. Therefore, these were added into the serum-free L-15 medium [80]. Cable EE and Isom HC [83] also reported that trace metals, such as copper, iron, and zinc, are required for the long-term proliferation of primary hepatocytes cultured in a chemically defined medium. They observed the growth of both smaller hepatocytes and typical MHs in the medium supplemented with 2% dimethylsulfoxide (DMSO) for several months. Isom HC et al. [84] reported that adult rat hepatocytes that were cultured in the medium supplemented with 2% DMSO survived longer and maintained the capability of albumin synthesis longer than when cultured in the commonly used serum-free medium. The addition of 2% DMSO to the culture medium inhibited most hepatocyte proliferation, while 1% DMSO did not inhibit hepatocyte proliferation but did inhibit nonparenchymal cell (NPC) proliferation, especially mesenchymal cells that include fibroblasts (Figure 1Bf) [9]. Fibroblasts (mesenchymal cells) limit the expansion of SH colonies (Figure 1Be), thus 1% DMSO was added to the medium from day 4 after plating [49].

The growth of SHs also depends on an extracellular matrix (ECM) protein that is used to coat the culture devices. SH colonies similarly expanded in dishes coated with collagen (Col) types I and IV and fibronectin (FN), but their expansion was faster in dishes coated with thin-Matrigel or Col-gel. In particular, the expansion of SH colonies was dramatically enhanced (approximately 1.4 times larger than without DMSO) when SHs were cultured on Col-gel in a medium supplemented with 1% DMSO (Figure 1Be,f) [85]. These results indicate that the culture environment can bring out the innate potential of individual hepatocytes. They also indicate that SHs have a very high proliferative capacity and that the extent of their proliferation depends on the composition and three-dimensional ECM structure.

### 4.2. Purification of SHs

We first used a simple method of low-speed centrifugation with sequential changes in gravity (50 and 150× *g*) and time (1 and 5 min) to isolate SHs [48,49]. This separation protocol contaminated NPCs, such as stellate cells, liver epithelial cells (LECs), Kupffer cells, and sinusoidal endothelial cells (SECs), as well as MHs, in the cell suspension. Additionally, both nicotinamide and fetal bovine serum (FBS) were required for seeded SHs to proliferate and form colonies [48]. The addition of FBS, as well as growth factors, was also necessary for SHs not only to proliferate but also to secrete plasma proteins such as albumin, transferrin, ceruloplasmin, etc. [9]. However, FBS can be removed from the culture because the proteins secreted by SHs can support their proliferation once SHs have increased in number [9].

SHs start cell division at 2–3 days after plating and proliferate clonally to form colonies when both nicotinamide and growth factors, such as EGF and HGF, are added to the medium [48]. Colony size varies among SHs, with some continuing to expand and others ceasing to grow after a relatively short incubation period. Most SECs disappear within a few days, and contaminated MHs divide two or three times at most. Most SHs can survive for more than a month but never achieve enough confluence to occupy the surface of the dish because many SHs in the colony suddenly die by apoptosis as the culture continues. However, SHs can be cultured for >5 months with repeated death and neogenesis [85]. Many SHs can spontaneously mature into cells with functions equivalent to those of MHs by interacting with NPCs (Figure 1Ab,e) [49,85] or by covering colonies with Engelbreth–Holm–Swam gels (EHS gels, Matrigel^®^, Growth-factor-reduced, Corning, NY, USA) [86].

Conversely, SHs can continue to proliferate on collagen gel without maturation when NPC proliferation is suppressed by the addition of 1% DMSO (Figure 1Bf), indicating that basement membrane reconstruction through the SH–NPC interaction is important for SHs to differentiate into MHs [49,86]. Additionally, primary hepatocytes have been thought not to proliferate after cryopreservation, but SHs can proliferate after long-term cryopreservation while maintaining their properties [87,88].

The standard method of preparing an SH-rich fraction uses multistep centrifugations and gives a mixture of SHs and NPCs, as described above. Therefore, we searched for specific surface markers of SHs by comparing SH-rich and MH-rich fractions using a DNA microarray for purifying SH. The comprehensive analysis identified CD44 as a molecule that is specifically expressed on the surface of SHs but not on MHs [89]. The *CD44* gene encodes for a family of alternatively spliced multifunctional molecules, and it plays a role in the adhesion of cells to ECMs such as hyaluronic acid (HA), collagens, and fibronectin [90]. SHs expressed CD44 standard type (CD44s) and variant type 6 (CD44v6) [89]. CD44s becomes positive by immunostaining from day 3 after plating, and CD44v6 expression is delayed compared to CD44s expression, but both CD44s and CD44v6 expression disappear once SHs mature. CD44-positive hepatocytes cannot be identified in normal rat liver, but they transiently appear in the periportal region when the rat liver is severely injured by hepatotoxins such as d-galactosamine (GalN) [89,91,92].

SHs that appeared in GalN-treated rat livers could be isolated using anti-CD44 antibodies. CD44 is not expressed on hepatocytes in the normal adult liver, but it is expressed in cultured SHs forming colonies. Hence, HA, the ligand for CD44, was thought to be able to selectively isolate SH populations. Among NPCs, only SECs express receptors for HA, LYVE-1, and stabilin-1/2 [93,94], but are CD44-negative. Most SECs die within a few days after plating, as mentioned above.

SHs selectively grow and form colonies in the serum-free DMEM/F12 medium when an SH fraction is plated on HA-coated dishes (Figure 1Af) [95]. The effect of HA does not differ among commercially available forms of HA, but the combination of nicotinamide and growth factors, as well as transferrin and selenium, are both required for cell growth in a serum-free culture medium. SHs form colonies that consist of 30–40 cells at 8–10 days after plating. SH colonies gradually detach more easily from the HA-coated dish after approximately 2 weeks of incubation, and SHs can be collected and cryopreserved without trypsin [95]. Long-term cryopreserved SHs could grow without losing their ability to proliferate and mature [88].

### 4.3. Self-Renewal of Hepatic Progenitor Cells

Most SH colonies uniformly consisted of SHs in early culture. Small mononuclear cells proliferated, while some colonies appeared to have MH-like characteristics with a large cytoplasm and a nucleus, sometimes with binuclei [9]. This finding indicates the existence of parental cells within a population of SHs, a genuine hepatocytic progenitor. Proving that SHs retain the ability to self-renew themselves generation after generation is necessary to identify the parental SH. Therefore, we first searched for conditions under which a single SH isolated from a colony could clonally proliferate [59].

SHs were cultured for 9–10 days in HA-coated dishes using a serum-free culture medium. Cells were detached using collagenase and hyaluronidase and then sorted using an anti-CD44 antibody to avoid trypsin injury. The sorted CD44-positive SHs failed to form colonies on either Col-I-coated or HA-coated dishes but proliferated to form colonies on thin-Matrigel-coated dishes. Furthermore, nicotinamide was crucial for colony formation, and the addition of EGF, insulin, and dexamethasone promoted parental SH growth [59]. However, a large number of the sorted cells seeded on thin-Matrigel-coated dishes failed to adhere and proliferate. The colonies demonstrated similar characteristics at the beginning of culture, but two morphologically distinguishable types of colonies emerged with time. One colony was circular and composed of small mononuclear cells, while the other demonstrated an irregular shape and was composed of cells with a large cytoplasm. The size of the colonies also varied. Colonies composed of small mononuclear cells, a typical SH, expanded rapidly to form large colonies, while colonies composed of relatively large cells proliferated slowly. This finding indicates the presence of a highly proliferative cell population. Only approximately 20% of the plated cells could adhere to a new dish that was thinly coated with Matrigel, which was small mononucleate cells when the cells were passaged every 4 weeks after the cultivation (Figure 1Ac). The SHs with high proliferative potential divided >50 times in 17 weeks. We defined the cells with high proliferative potential as hepatocytic parental progenitor cells (HPPCs) [59].

The HPPCs that grew on dishes coated with thin-Matrigel indicated that a certain component of Matrigel is crucial for HPPCs to maintain their ability to self-renew. Matrigel contains components that make up the basement membrane [96], which consists of laminin (LN), Col-IV, nidogen, and heparan sulfate proteoglycans [97]. Among them, LN is the major adhesion protein and mediates cell adhesion to the basement membrane. LNs are composed of three polypeptide chains, designated as α, β, and γ, and five α(α1–5), three β(β1–3), and three γ(γ1–3) chains are recognized in mammals [98]. Matrigel contains LN111 (α1,β1,γ1) as major constituents [96]. Generally, the LNα1-chain is expressed in fetal and neonatal rat liver lobules but is not found in adults. Instead, the LNα5-chain is present exclusively in the Glisson sheath [99]. Conversely, the transient expression of LNα1 was observed in regenerating liver lobules after 2/3 PH [99]. Integrins play a crucial role in cell adhesion to LNs [98]. They are composed of noncovalently associated α and β subunits. At least 24 separate integrins consisting of distinct combinations of α and β subunits have been identified in mammals to date. The specificity of the LN–integrin interaction is mainly dependent on α chains of LN, and the ligand specificity of the integrin is primarily determined by the α subunits. Integrin β subunits play a crucial role in signal transduction but have an auxiliary role in the ligand specificity [97,98].

Only <20% of SHs could adhere to dishes that were coated with LN111, as observed in thin-Matrigel-coated dishes [100]. The cell adhesion rates were slightly lower on LN111 than on Matrigel, and only <20% of cells adhered to the new dishes when SHs grown on LN111 were passaged. This rate gradually decreased with each successive passage. This indicates that the number of HPPCs decreased with passages. Furthermore, many of the cells that failed to attach to LN111 attached to LN511, indicating that HPPCs do not appear on LN511 and that cells growing on LN511 have lost their self-renewability. Thus, HPPCs generate two distinctive cell populations, which may indicate that HPPCs perform asymmetric cell divisions: LNα1-dependent and LNα5-dependent. HPPCs’ self-renewability is LN111-dependent, whereas LNα5, which is produced by adherent HPPCs, may support the survival and proliferation of LNα5-dependent daughter cells. Among the integrins that are involved in LN-binding, integrins α3 and β1 were expressed more in SHs that proliferate on LN111 than in cells on LN511, while integrin β4 was more strongly expressed in cells growing on LN511 [100]. Integrin α3^high^α6β1^high^ could form HPPC colonies on LN111, but not α6β1^low^ cells. Neutralizing antibodies against LN111 and integrin β1 could inhibit HPPC colony formation on LN111. These results indicate the importance of signaling from LN111 via integrin β1 for HPPC proliferation.

LNα1 and LNα5 were detected immunohistochemically on the basolateral side of cholangiocytes in fetal mouse liver, whereas only LNα5 was found in the adult liver [101]. Maintaining the self-renewability of LN111 has been reported in hepatoblast-like cells derived from human-induced pluripotent stem cells (hiPSCs) [102]. Conversely, undifferentiated hiPSCs can be maintained on LN511 but not on LN111 [103]. LN111 not only selectively maintains hepatoblast-like cells, but also eliminates the remaining undifferentiated cells. Additionally, CD45^−^TER119^−^c-Kit^−^c-Met^+^CD29^+^CD49f^+/low^ cells [104] and CD45^−^TER119^−^Dlk^+^ cells [105] have been reported to be highly proliferative and bipotential in fetal mouse livers. Furthermore, CD45^−^TER119^−^Dlk^+^ cells were named hepatic progenitor cells that proliferate on LN (HPPLs) because they can maintain high proliferative potential on LN-coated dishes [106]. HPPLs are Dlk^−^Krt19^+^albumin^+^, whereas hepatoblasts are Dlk^+^Krt19^−^albumin^+^, indicating that HPPLs lose the characteristics of hepatoblasts. However, HPPLs possess not only high proliferative capability but also the potential to differentiate into both hepatocytes and cholangiocytes. LN is a crucial factor in maintaining the proliferation and multilineage differentiation potential of HPPLs. Furthermore, HPPLs strongly expressed integrin α6β1, and signals from LN are transduced through integrin β1 [106]. Conversely, intercellular cell adhesion molecule (ICAM)-1-positive liver progenitor cells separated from late fetal and postnatal mouse livers maintained their self-renewability on LN111 after passaging [53]. Furthermore, RT1A^1−^OX18^low^ICAM-1^+^ cells isolated from E13 rat livers have been reported to express integrin β1, proliferate on LN, and maintain bipotentiality [107]. Additionally, RT1A1^+^OX18^+^ICAM-1^+^ hepatocytes isolated from adult rat livers have also been reported to have a highly proliferative capability, but these cells must be cultured in a serum-free defined medium that contains EGF using STO embryonic cell lines as feeder cells. Thus, the self-renewability of cells with hepatocytic features may be maintained in the LNα1-dependent manner.

### 4.4. Characteristics of HPPCs

Gene expression patterns are also different between cells on LN111 and LN511. The gene expression levels of *Alb*, *carbamoylphosphate synthetase* (*Cps*) 1, *glutamine synthetase* (*Gs*), *Keratins* (*Krts*) *8 and 18*, *hepatocyte nuclear factor* (HNF) *4α* (*Hnf4α*), and *CCAAT/enhancer binding protein* (C/EBP) *α* (*Cebpα*), as well as *Cd44*, *a-fetoprotein* (*Afp*), and *Dlk-1*, were found in both LN111-dependent and LN511-dependent cells [100]. However, absolute values of the genes related to hepatic functions, such as *Alb*, *Cps1*, *Hnf4α*, and *Cebpα*, were much lower in HPPCs than in MHs. Cholangiocyte-related genes, such as *Krts 7 and 19*, *Sox9*, and *epithelial cell adhesion molecule* (*Epcam*), were more highly expressed in cells cultured on LN511 than on LN111. Conversely, hepatic stem-cell-related genes, such as *Thy1*, *c-Kit*, *Ncam1*, and *Cd24*, were more highly expressed in cells cultured on LN111 than on LN511. Additionally, the genes of *Cd34*, *leucine-rich orphan G-protein-coupled receptors* (*Lgrs*) 4 and 5, *Axin2*, and *telomerase reverse transcriptase* (*Tert*), were expressed in cells neither in LN111 nor in LN511 [100]. Recent lineage tracing studies in mouse liver have revealed the presence of cell populations that regenerate the liver under homeostatic or injury conditions. Periportal hepatocytes that express *Sox9* [108] or *Mfsd2a* [109], pericentral *Axin2^+^* [110,111] or *Lgr5^+^* hepatocytes [112], and broadly distributed *Tert^high^* [113] or *Lgr4^+^* hepatocytes [114] have been identified as candidates for generating new hepatocytes in homeostasis although they are strongly debated [115,116,117,118,119,120,121]. However, our results indicate that HPPCs may be different from those subpopulations.

## 5. SH in Liver Lobules

SHs emerge not only in in vitro but also in vivo in rodents and human livers. Isolated hepatocytes are smaller in size than typical MHs, of which morphology is closer to SHs, before weaning. The frequency of the small-sized hepatocytes dramatically decreased with age after weaning [38]. Sigal SH et al. [122] used fluorescence-activated cell sorting (FACS) to reveal that hepatocytes that were isolated from fetal and suckling rat livers were all mononuclear and possessed granularity and autofluorescence comparable to hepatoblasts. The presence of small-sized hepatocytes has also been discovered in adult rat livers. Tateno C et al. [123] revealed that the isolated hepatocytes from male F344 rat livers based on the granularity and the autofluorescence by FACS were categorized into three subpopulations: SH-R3 (17.1 ± 0.2 μm), SH-R2 (22.6 ± 0.5 μm), and PH (24.1 ± 0.5 μm). Subsequently, Asahina K et al. [124] showed that >80% of SH-R3 cells were mononucleate and diploid and possessed high replicative capability.

Conversely, the administration of hepatotoxins, such as [*5.1*] 2-acetylaminofluorene (2-AAF), [*5.2*] d-galactosamine (GalN), and [*5.3*] retrorsine (Ret), induce the appearance of small-sized cells in rodent livers with or without growth stimulation such as PH. [*5.4*] Intermediate hepatocytes (IHs) have been observed in human livers with acute or chronic liver diseases. Here, we introduce the “small-sized hepatocytes” that appear in the liver, based on the literature and our experience. Additionally, Table 1 shows the characteristics of the related cells.

### 5.1. The 2-AAF/PH Model

Generally, 2-AAF interacts directly with DNA in hepatocytes to produce DNA adducts. It is used to induce a complete blockage of the proliferative capacity of hepatocytes after PH. This 2-AAF/PH protocol has been used as the most reliable rat OC model [10,135,136,137,138]. However, it cannot be applied to the mouse experiment because the mouse liver lacks the enzyme N-sulfotransferase, which activates 2-AAF [139]. OCs are known to express markers for membrane proteins, such as CD34, c-Kit, and Thy1, which are also regarded as hematopoietic stem cell markers [17]. Therefore, OCs were thought to be derived from BM [140]. Conversely, Thy1 has been reported to be a marker of hepatic myofibroblasts and hepatic stellate cells rather than specific for OCs [141,142]. Whether Thy1 is a specific marker for OCs remains controversial, but we have reported that Thy1^+^ cells that are isolated from GalN-treated rat liver contain an LSPC (liver stem/progenitor cell) population [90,91].

Two patterns in the differentiation process of OCs into hepatocytes were reported depending on 2-AAF dosage [10,135]. The basic difference between the two doses was recognized a few days after PH. They revealed that OCs were arranged in a straight tubular pattern in rat livers that were treated with a low 2-AAF dosage, and most OCs synchronously differentiated into basophilic SHs (bSHs). The few unchanged OCs were located on the periportal side of the duct, whereas the distal part of the same duct was composed of bSHs. The newly formed hepatocytes maintained their tubular structure during the early stages of the differentiation process. Recovering normal lobular structure takes 10–12 days after PH. In contrast, foci that consist of bSHs appeared in livers treated with high doses of 2-AAF, where the basement membrane is histologically undetectable. Newly formed hepatocytes consisted of foci scattered throughout the livers, not confined in the periportal regions. Reconstructing the liver structure after PH may take approximately a month. The hepatocytic differentiation process of OCs was correlated with HNF4α expression and basement membrane disappearance [10]. Thus, the phenotypes were altered from a hybrid type of cells that express markers of hepatoblasts and BECs (AFP, OV-6, and Cx 43) to hepatocytes (HNF4α, α1-AT, Cx32, and BC formation) during OC differentiation into hepatocytes.

### 5.2. d-Galactosamine (GalN) Model

GalN is a potent hepatotoxic agent, causing hepatocyte death both by necrosis and apoptosis. GalN is metabolized by hepatocytes mainly in the pericentral region and functions as a transcriptional inhibitor by depleting uridine nucleotides [143,144]. Three unusual epithelial cell types, including OCs, bSHs (<16 μm), and hepatocytes, appear on tubular structures between 3–5 days after the intraperitoneal administration of GalN to rats (Figure 1Cb,d,e) [101,145,146]. The number of bSHs reaches a maximum on day 4 after GalN administration and then decreases. The liver finally regains its normal structure by day 8. We have reported the appearance of SHs to be different from bSHs in GalN-injured livers [89]. Thy1-positive cells emerged in the periportal region on day 2 when rats were treated with GalN (Figure 3A). CD44-positive SHs transiently appeared in the region between Thy1^+^ cells and viable hepatocytes on days 3–5 after GalN treatment after the appearance of Thy1-positive cells. The number of CD44-positive SHs reached a maximum on day 4, while Thy1-positive cells rapidly declined and almost disappeared from the lobules by day 4. Thereafter, the number of CD44-positive SHs declined rapidly, and finding the cells that expressed CD44 was difficult on day 6.

The isolation of the SH fraction, which contains HSCs, SECs, Kupffer cells (KCs), and LECs as well as SHs, yielded approximately 1 × 10^8^ cells from the liver of an adult male F344 rat at day 3 after GalN treatment (GalN-D3). The cells were then divided into Thy1^+^ and CD44^+^ fractions using anti-Thy1 and anti-CD44 antibodies, respectively. Thy1-positive cells that were isolated from GalN-D2 mostly demonstrated the characteristics of fibroblasts, while Thy1-positive cells that were isolated from GalN-D3 contained both morphologically polygonal (epithelial-like) and spindle shapes (fibroblast-like) [91,92]. Approximately 45% of fibroblasts predominantly showed Thy1^+^/desmin^+^ whereas approximately 55% of epithelial-like cells showed Thy1^+^/Krt19^+^, Thy1^+^/albumin^+^, or Thy1^+^/CD44^+^. In contrast, most of the CD44-positive cells that were isolated from GalN-D3 were polygonal in shape, and their morphology characterized the epithelial cells. Sorted CD44-positive cells expressed Krt19, albumin, and Thy1 at approximately 60%, 62%, and 65%, respectively [92]. Furthermore, approximately 2.0% ± 0.5%, 3.1% ± 0.3%, and 3.0% ± 0.6% of the SH fraction were Thy1^+^/CD44^−^, Thy1^+^/CD44^+^, and Thy1^−^/CD44^+^ cells, respectively (our unpublished data). The size and the characteristics of CD44^+^/Krt19^+^ cells were similar to those of the Thy1^+^/Krt19^+^ ones. Conversely, the colony formation efficiency of Thy1^−^/CD44^+^ cells was five times higher than that of the Thy1^+^/CD44^−^ and Thy1^+^/CD44^+^ ones. Additionally, the average size of colonies composed of CD44^+^/albumin^+^ cells was larger than that of the colony composed of the Thy1^+^/albumin^+^ ones. Moreover, the average sizes of colonies derived from CD44^+^ cells (GalN-D4) at day 10 after plating were approximately twice the average size of colonies derived from the CD44^+^ ones (GalN-D3) [91].

EGF, HGF, and basic fibroblast growth factor (bFGF) must be added to the culture medium alone or in combination to induce CD44^+^ SH colonies from the Thy1^+^ cells of GalN-D2 [91]. Conversely, Thy1^+^ cells that were isolated from GalN-D2 could not differentiate into BECs even in collagen-sandwich culture using the BEC-induction medium, while Thy1^+^ (GalN-D3) cells could form cord- and/or cyst-like structures. These results indicate that cells, among the Thy1-positive cells isolated from GalN-D3, have acquired the bipotential ability to differentiate into either hepatocytes or BECs. Furthermore, cells that have lost Thy1 expression failed to differentiate into BECs [92]. Comprehensive analysis of gene expressions confirmed that Thy1-positive cells differentiate into hepatocytes, and Thy1-positive epithelial-like cells that appeared in the periportal region on day 2 after GalN treatment may differentiate into Thy1^+^/CD44^−^, Thy1^+^/CD44^+^, Thy1^−^/CD44^+^ SHs, and finally Thy1^−^/CD44^−^/C/EBPα^+^ (MHs) cells, in that order (Figure 3B) [91,92]. Transplantation of Thy1^+^ cells into Ret/PH-treated rat livers verified that the cells possessed bipotentiality [147]. Some donor cells were incorporated into the hepatic cords and differentiated into hepatocytes, with a small number of bile ductules composed of Thy1^+^ cells derived from donor cells. The relationship between hepatic differentiation and loss of Thy1 expression has also been reported in fetal livers [148,149,150,151]. A small number of AFP^+^Alb^+^Krt19^+^Ecad^+^ were in the Thy1^+^ cell fraction isolated from rat E14 fetal liver, while most cells that expressed hepatic markers (AFP^+^Alb^+^Krt19^+^) were in the Thy1-negative fetal liver cell fraction [151].

CD44-positive cells are hepatocytic progenitors, thus CD44-positive cells with SH-colony-forming ability that appear in GalN-injured rat livers are thought to have originated from two distinctly different cells: OCs and MHs. In particular, the newly arising hepatocytes are derived from both the remaining MHs and OCs, but the majority of them are thought to be MH-derived cells since OC proliferation is mild in GalN-injured livers (Figure 4).

### 5.3. Retrorsine/PH Model

Ret is a member of the pyrrolizidine alkaloid family of naturally occurring compounds that are toxic to various mammalian tissues [152]. The hepatotoxic effects of Ret are long-lasting, and systemic administration of Ret severely inhibits the proliferative ability of MHs. Ret-treated hepatocytes can synthesize DNA but are unable to complete cell division when the liver is subjected to a strong proliferative stimulus such as PH or massive hepatocellular necrosis. Hence, non-proliferating giant hepatocytes (megalocytes) are formed [21,153,154,155,156]. Hepatocytes were unable to proliferate in the liver of rats treated with Ret/PH, thus small-sized hepatocytes appeared instead, actively proliferating in clusters within the liver lobule [153]. The cluster-forming cells were considered endogenous hepatocyte progenitor cells and were named small hepatocyte-like progenitor cells (SHPCs) [21]. SHPCs are morphologically distinct from the surrounding hepatocytes and are observed from approximately 3 days post-PH, proliferating and forming clusters (Figure 1Ca). SHPC clusters are found in all lobular zones (31%, 43%, and 26% in zones 1, 2, and 3, respectively) [21]. We examined the location of SHPC clusters at 14 days post-PH by measuring their distance from CV and PV and revealed that approximately 55% and 45% of the clusters were localized in zones 1 and 2, respectively. However, no clusters were found in zone 3 (our unpublished data). Conversely, Gordon GJ et al. [157] reported that >90% of cells forming clusters were positive for Ki-67 at 14 days post-PH, while <20% of SHPCs were Ki-67^+^ in our experiments [158]. SHPC proliferation was reduced and liver mass was fully restored by 30 days after PH [21]. TUNEL-positive nuclei are often found in the Ret-damaged MHs, typically in megalocytes surrounding SHPC clusters, but rarely in SHPCs. The number of apoptotic cells in Ret-treated livers peaked at 1 day after PH (approximately 6%) and then decreased [157]. OC proliferation was moderate in this model, peaking 7 days after PH [21]. The emergence and expansion of SHPCs may compensate for the lost liver mass by PH instead of MHs receiving the growth suppression by Ret, considering that OC differentiation into bSH was rarely observed. Thus, the Ret/PH-treated livers regained their original mass after approximately 1 month, whereas healthy rat livers recovered in 7–10 days (Figure 5).

The cells of origin of SHPCs remained unknown and hotly debated [159,160]. Several possible cells of origin for SHPCs have been indicated, including OCs [161,162], retrorsine-resistant hepatocytes [163,164], and a novel progenitor cell population in the lobules [21,165]. The morphological features of SHPCs observed early after PH are similar to those of MH, except for size [21]. SHPCs share some phenotypes in common with hepatoblasts, OCs, and MHs, making them more like SHs than MHs. A subset of SHPCs express the OC/BEC/hepatoblast markers (OC.2 and OC.5) through 5 days after PH [156,166]. Gene and protein expression analyses in the earliest SHPCs revealed that all of the major liver-enriched transcription factors, Wilms’ tumor 1, AFP, and P-glycoprotein were expressed, whereas the expressions of tyrosine aminotransferase and α1-AT were reduced compared to surrounding MHs [166]. Cytochrome P450 (CYP) 2E1 and CYP3A1 in the rat liver are known to be undetectable until near or at birth [167,168,169]. Ret administration to rat livers has induced *Cyp2e1* and *Cyp3a1* expression [170], but those expressions were absent [166]. Our comprehensive analysis of *SHPC* gene expression revealed that many genes related to differentiation functions returned to the MH levels around the SHPC cluster, except for *Cyp2b1*, in rat liver 14 days after PH [171]. The lack of significant CYP reduction may give SHPCs resistance to the mito-inhibitory effect of Ret, which is required to metabolize Ret to its toxic derivative. SHPCs have reduced the expression of many *CYP* genes and the differentiation-function-related genes compared to MHs [171]; thus, they are considered hepatocyte-derived and SH-equivalent cells, not OC-derived cells. Additionally, a novel progenitor cell population, as proposed by Gordon GJ et al. [21], is unlikely to exist within the Ret/PH-treated lobules (Figure 5).

Examining whether isolated SHPCs could proliferate in culture is important. Gordon GJ et al. [172] revealed that isolated SHPCs from Ret/PH-treated rat livers at 6–8 days and 13–15 days post-PH did not proliferate or form colonies. We also examined whether isolated SHs from Ret-treated rat livers could proliferate and revealed that no colonies formed when SHPCs were isolated and cultured from Ret-treated livers, but many SH colonies were formed when isolated cells immediately after PH from Ret-treated livers were plated onto thin-Matrigel- or LN-coated dishes (our unpublished data). These results indicate that the cells must be exposed to growth stimuli in vivo for SHPCs to proliferate in vitro.

### 5.4. Appearance of Intermediate Cells in the Human Liver

Generally, small-sized hepatocytes are present in the human liver. These cells exhibit traits that are intermediate between MHs and BECs. Intermediate hepatobiliary cells (IHBCs) were defined as >6 microns in diameter (the approximate size of normal CoH cells, i.e., the smallest cholangiocytes) but <40 microns (the typical size of a hepatocyte) at a consensus meeting of researchers in liver diseases, with other features indicating dual characteristics of both MHs and BECs [22]. IHBCs are predominantly found in the liver with moderate to severe inflammation. The number of IHBCs gradually increased as inflammation progressed to more severe levels and hepatocyte necrosis expanded in the advanced stages of necrotizing hepatitis and (non)alcoholic steatohepatitis [173,174,175]. The most widely used markers of IHBC are anti-Krt antibodies that target biliary-type keratins (Krt7 and Krt19), and Krt7-positive hepatocytes are recognized in liver cell rosettes and even as single cells or subpopulations distributed heterogeneously throughout the liver lobule [176]. Desmet VJ [126] revealed that the differentiation stage of IHBCs can be recognized by the expression gradient of Krt7 and Krt19, and the differentiation of LSPCs into hepatocytes can be found from Krt19^+^ LSPCs to Krt19^−^Krt7^+^ IHBCs, and finally Krt 7^−^ hepatocytes (ductular hepatopoiesis). In contrast, MH dedifferentiation progresses from Krt7^−^ hepatocytes through Krt7^+^ IHBCs, and finally, to Krt19^+^Krt7^−^ ductular reaction (DRs). Rodent OCs and human IHBCs share important physiological roles despite the marked morphological and phenotypic differences. They simultaneously express biliary antigens (Krt7, Krt19, and OV-6) and hepatocyte antigens (HepPar1, albumin, α1-AT, and sometimes AFP) [22].

Massive hepatic necrosis (MHN) is associated with fulminant hepatic failure (FHF) and is a rare but very serious complication caused by a variety of etiologies [177]. This condition frequently causes death in patients, but 10–20% are known to recover spontaneously without liver transplantation [131]. MHN in FHF exhibits unique pathophysiologic features, including rapid hepatocyte death and regeneration. The initial regenerative response in FHF is primarily caused by surviving hepatocytes. LSPCs begin to proliferate vigorously to compensate for the massive loss of hepatocytes when the injury persists or most hepatocytes are impaired [178].

The most commonly recognized tissue reaction is the appearance of a tubular response called DR, which corroborates the presence of LSPCs in the human liver. The degree of LSPC activation in acute or subacute liver injury correlates with histopathologic and clinical disease severity [27,179]. LSPC activation and differentiation were more prominent in areas with more severe hepatocyte loss than in areas with less severe hepatocyte loss, and a threshold of 50% loss of hepatocytes (marked reduction in proliferative potential of remaining hepatocytes) was required for extensive LSPC activation. This finding is similar to the dose-dependent differentiation of rat OCs that appeared after 2-AAF treatment [10]. Thus, IHs were already fully differentiated in surviving patients, while this differentiation process (LSPCs-IHs-MHs) was inhibited in deceased patients [179].

The IH-mediated differentiation process from LSPCs to MHs is observed in the livers of patients who receive auxiliary partial orthotopic liver transplantation (APOLT) [180]. APOLT is unique in that the damaged liver is left in situ to allow for regeneration even after severe damage to a degree not compatible with life [181]. DR activation was observed in the injured liver, and DRs always emerge from the periportal region and extend from there, and their intensity correlates with the degree of hepatocyte injury. Ductular cells demonstrated an intermediate cell type with both biliary (Krt7/19) and hepatocyte markers (HepPar 1, α1-AT, and HNF4α), indicating that DRs are a cell population originating from LSPCs and play an important role in liver regeneration. This notion was supported by sequential biopsies of the native liver of a patient who received APOLT [180].

The intensity of DRs gradually increased, after which round clusters of hepatocyte-like cells were formed, and finally, the trabecular pattern of hepatic parenchyma was re-established. Quaglia A et al. [182] revealed that hepatocyte proliferation occurred in the native liver within a few days after transplantation in patients undergoing APOLT for acute liver failure with a diffuse pattern of liver injury. Hepatocellular proliferation and potential ductular hepatopoiesis were variably involved in the regeneration when the injured area was mapped. These results indicate that LSPCs can proliferate and differentiate into MHs to replace lost hepatocytes if liver function is adequately supported by the transplanted liver. LSPC and DR activation are also observed in human chronic liver disease. Cells that share characteristics of hepatocytes and BECs appear in DRs and these cells are termed “ductular hepatocytes (DHs)” [27,183,184,185]. These duct-like structures frequently have a biliary appearance at one end and a hepatocytic appearance at the other end, of which cells demonstrate a range of morphologically and immunohistochemically intermediate phenotypes [129,134]. DHs are considered to originate from the CoH [186].

A general trigger for the activation of LSPCs is the inability of MHs to activate, as seen in experimental rodent models. The proliferative capacity of MHs is lost due to replicative senescence caused by telomere shortening in the cirrhotic stage of a wide variety of chronic human liver diseases, and replicative senescence of MHs is thought to be caused by progressive proliferation over the 20–30 years of chronic liver disease [187,188]. The entire regenerative nodules in cirrhosis are known to be composed of IHs, which are clonal [113,189] and may originate from LSPCs [175,190]. Thus, these results indicate that the transient appearance of SHs, called IHBCs, IHs, or DHs, is always observed during the regenerative process of severely damaged livers due to various diseases.

## 6. Growth Regulation of Small Hepatocytes

SH proliferation is known to be regulated by a variety of factors (Table 2). Growth factors, such as EGF, HGF, and/or TGF-α, are required to induce SH colony formation in primary cultures [48,191]. The ability of EGF, HGF, and/or TGF-α to induce colony formation is almost identical, and their combination did not act synergistically on SH emergence or SH proliferation. FGF could induce SH colony formation, but its colony formation efficiency was not as high as that of EGF, HGF, and TGF-α [191]. SHs require continuous growth stimulation to proliferate and form colonies in the early culture period.

Agents that affect the proliferation of primary hepatocytes fall into two groups [197]. One is the complete mitogen, which can stimulate DNA synthesis and mitosis by itself in chemically defined serum-free media. The other is a co-mitogen that coexists with the complete mitogen to promote proliferation. Complete mitogens include EGF, HGF, TGF-α, and FGF. Co-mitogens include tumor necrosis factor (TNF)-α, insulin-like growth factor (IGF)-I, norepinephrine, angiotensin-II, vasopressin, etc. [197]. The complete mitogens and pleiotrophin [192] can induce SH appearance and proliferation, while co-mitogens cannot induce SH appearance by themselves, as mentioned above. However, they can enhance the ability of complete mitogen to increase the number of SH colonies [191]. IL-6 inhibited the proliferation of primary rat hepatocytes [195], but IL-6 and stem cell factor (SCF) promoted SH colony formation in a medium that contains EGF, and the combination of IL-6 and SCF significantly increased the number and size of SH colonies [171].

Conversely, the administration of dexamethasone (Dex) before PH has been reported to inhibit the appearance of SHPCs, resulting in the suppression of liver regeneration and significantly increased short-term mortality in the liver of Ret/PH-treated rats [193]. However, IL-6 treatment restored liver regeneration and reduced mortality in Dex-treated Ret/PH rats, and the number of expanded SHPCs increased in IL-6-treated livers compared to livers not treated with IL-6. Considering that Dex administration can inhibit TNF-α and IL-6 production by interfering with the transcription of their respective genes [198,199,200], these results indicate that cytokine priming is required for the appearance of SHPCs in Ret/PH-treated rat liver and that IL-6 directly mediates the SHPC response [193]. The effect of Dex on SH colony formation may be difficult to assess in vitro because the addition of insulin and Dex to the culture medium is indispensable for primary hepatocytes to survive and be maintained [9].

Generally, the activin (Act)/follistatin (Fst) system crucially contributes to cell proliferation homeostasis in the normal liver [201,202,203,204]. MHs constitutively produce ActA, which acts as a hepatocyte growth suppressor in autocrine/paracrine [205]. Additionally, ActA expression was reported to be increased in aged rat liver [67]. We reported that SH secretes ActB and Fst into the culture medium and the secretion of both factors increases during SH proliferation [196]. The bioactivity of Act is strongly inhibited when bound to Fst which acts as an extracellular binding protein [206]. Consistently, Fst administration after PH inhibits the action of Act and consequently promotes liver regeneration [207]. Conversely, Fst expression may be important for SH proliferation. Secreted Fst may stimulate SH proliferation, possibly by inhibiting ActB produced by SHs in an autocrine manner [196]. Exogenous administration of ActA or ActB to SH culture medium failed to induce Fst expression in SHs. Fst expression decreased while ActA expression increased, and SH growth was inhibited when SHs matured spontaneously over the course of the culture. Further, ActB is known to have a similar physiological activity as ActA but has less activity than ActA [208]. We also confirmed that ActB is less potent than ActA in inhibiting SH proliferation [196]. Furthermore, ActA administration is known to induce the apoptosis of MHs [201,209]. SH proliferation was strongly inhibited by ActA and ActB, but neither ActA nor ActB administration induced apoptosis [196].

Differences between MHs and SHs in the Act signaling pathways may have caused this discrepancy. Similar to CD44-positive cancer stem cells that are resistant to apoptotic signaling [210,211], OCs and LSPCs are resistant to TGFβ-induced apoptosis [212,213]. Smads are an essential component for TGFβ/Act signaling. Smad6 expression did not differ between SHs and MHs, but Smad7 expression was higher in SHs than in MHs [196]. Therefore, we hypothesized that CD44 and Smad7 expression in SHs may be responsible for their resistance to Act-induced apoptosis.

We have reported that cell transplantation into Ret/PH-treated rat livers affects SHPC proliferation [158,171,194]. Isolated Thy1-positive mesenchymal cells (hepatic Thy1) from GalN-treated rat liver and rat BM-MCs could stimulate liver regeneration of the recipients as well as hepatomegaly due to SHPC expansion. The number and size of SHPC clusters in the livers transplanted with hepatic Thy1 were significantly larger than those in control livers. Hepatic Thy1 consists of two types of cells, fibroblast-like mesenchymal cells (Thy1-MCs) and epithelial-like cells, but only Thy1-MCs can stimulate SHPC proliferation, while the latter cannot. However, epithelial-like cells contain LSPCs that differentiate into SHs [158,194].

Conversely, isolated Thy1-MCs from the BM of healthy rats also could promote SHPC proliferation [171]. Most Thy1^+^ donor cells failed to engraft and only a few cells remained in the recipient’s liver for a short period, but the proliferation-promoting effect of SHPCs persisted for >1 month. We speculated that donor cells that are caught in the sinusoids secrete extracellular vesicles (EVs), which in turn affect neighboring hepatocytes. EVs collected from cultured Thy1-MCs were administered to Ret/PH-treated livers and induced SHPC cluster expansion [158,171]. However, the target cells of EVs produced by hepatic Thy1 were different from the target cells of EVs by BM-MC.

Comprehensive analysis of SHPCs revealed that hepatic Thy1 transplantation increased *Il6r* and *Egfr* as well as *Il17rb* expressions [158]. *Il17b* expression was upregulated in SECs that were isolated from Ret/PH-treated livers, whereas *Il25* was expressed in KCs that were isolated from those with hepatic Thy1 cells. Neither *Il17b* nor *Il25* were expressed in SECs and KCs that were isolated from healthy livers. Hepatic Thy1-EVs enhanced *Il17rb* expression in cultured SHs, and SH proliferation was promoted by IL-17B and IL-25 administration, but the synergistic effect of IL-17B and IL-25 was not observed [158].

In addition, activated KCs by transplanted hepatic Thy1^+^ cells produced cytokine-induced neutrophil chemoattractant-2 (CINC-2) and *miR-199a-5p*, which promoted *Il17b* expression in SECs and SHPC proliferation, respectively. Furthermore, the conditioned medium of SECs treated with CINC-2 promoted SH proliferation [194]. These results indicate that hepatic Thy1 transplantation into Ret/PH-treated livers promotes SHPC proliferation via the IL-17RB signaling pathway and that *miR-199a-5p* may act directly on SHPCs to promote proliferation. Conversely, a comprehensive analysis of BM-MC-EVs revealed that they contain *miR-146a-5p*, IL6, and SCF, which may stimulate SH proliferation [171].

Gadolinium chloride (GdCl_3_) is known to inhibit the phagocytic function of KCs. Hepatic Thy1 transplantation into Ret/PH-treated rat livers that were treated with GdCl_3_ 24 h before PH markedly suppressed the expansion of SHPC clusters in the liver [171]. In contrast, BM-MC transplantation into Ret/PH/GdCl_3_ rat livers significantly increased the number and area of SHPC clusters. *Il17rb* expression in SHPCs was more strongly upregulated in livers transplanted with either BM-MCs or hepatic Thy1, but expression tended to be higher in SHPCs in livers transplanted with hepatic Thy1. These results indicate that SHPC expansion by hepatic Thy1 transplantation is dependent on the phagocytic activity of KCs. KC inactivation in livers transplanted with hepatic Thy1 resulted in the inability of CINC-2 and miR-199a secretion and the loss of growth stimulation via IL-17RB signaling to SHPCs. Conversely, KC inactivation in livers transplanted with BM-MCs allows donor cells to escape phagocytosis by KCs and remain for a relatively long time in the sinusoids. Figure 6 illustrates the interactions between transplanted Thy1^+^ cells and resident liver cells. Consequently, BM-MCs increased the secretion of both EVs and cytokines, which may give paracrine effects to neighboring cells [171]. Indeed, hepatic Thy1-EV administration to Ret/PH-treated livers did not result in the SHPC cluster emergence in zone 3, but approximately 10% of SHPC clusters were found in zone 3 in livers from rats transplanted with hepatic Thy1 (our unpublished data). In contrast, the distribution of SHPC clusters was not affected and few were observed in zone 3 in livers transplanted with BM-MCs or treated with BM-MC-EVs.

## 7. Participation of SHs in Liver Regeneration

Considering the knowledge accumulated to date on SHs, the involvement of SHs in liver regeneration can be summarized as follows.

### 7.1. Simple Loss of MHs

After 2/3 PH, the remaining hepatocytes are nearly intact and can regain lost volume by the cell division of the remaining hepatocytes. Miyaoka Y et al. [214] revealed that hepatocyte hypertrophy and proliferation contribute almost equally to regeneration after 2/3 PH. Only approximately half of the hepatocytes underwent cell division and binuclear hepatocytes divided into two mononuclear cells without DNA synthesis. Carbon tetrachloride administration caused liver regeneration similar to that of 2/3 PH. Both MH-I and MH-II divided equally until the original liver volume was restored in both cases because most hepatocytes synthesize DNA. Thus, LSPCs do not contribute to this regenerative process, and the clonal proliferation of MH-I (clusters of SHs) was not observed.

### 7.2. Growth Suppression of MHs

The process of liver regeneration may be different from a simple loss of hepatocytes when the proliferative capability of MHs is suppressed or the number of functional hepatocytes is far less to maintain homeostasis. The suppression of MH proliferation had three possible patterns: (1) when only MH-II is suppressed and MH-I divides; (2) when both MH-I and MH-II are suppressed; and (3) intermediate conditions are observed, and GalN treatment is a typical case. The timing of LSPC appearance may depend on the intensity of growth inhibition on hepatocytes.

(1)MH-II is sensitive to Ret, while MH-I is resistant in the case of Ret, as described above. Ret inhibits MH-II proliferation, but it does not affect hepatocytic differentiation function. Therefore, the basic hepatic function can be maintained by residual cells even after 2/3 PH in Ret-treated rats. MH-I actively proliferates to compensate for the lost cells instead of MH-II, resulting in the appearance of SHPCs, to restore the original volume (Figure 5). Therefore, it takes more time to recover the original mass in the Ret-treated animals compared to the untreated ones. Thus, generating new hepatocytes by inducing OCs is unnecessary.(2)Generally, 2-AAF completely inhibits hepatocyte proliferation after 2/3 PH. This indicates that both MH-I and MH-II cannot proliferate and the lost cells cannot be recovered by the remaining hepatocytes. Therefore, to restore reduced liver function, a strong growth stimulation drives OCs to expand and differentiate into hepatocytes. Increased numbers of OCs are arranged to create a small lumen, and a small number of OCs exist at the proximal end of the ductules (portal side). Cells at the distal portion of the same ductule differentiate into bSHs. Conversely, cell clusters composed of bSHs appear within the lobules in livers treated with high doses of 2-AAF.(3)The regenerative reaction of the liver after GalN administration is totally different from that after Ret or 2-AAF administration. Pericentral hepatocytes die while surviving hepatocytes are also weakly damaged. Therefore, the growth stimulus after injury is first applied to the bile ducts (CoH), causing the appearance of OCs (Thy1^+^/Krt19^+^/CD44^−^ cells) that form small ductules. The growth inhibition of MH-I is weaker than that of MH-II, causing the appearance of CD44^+^ hepatocytes with a slightly delayed response of MH-I to the growth stimulus. Approximately 3 days after GalN treatment, Thy1^+^/CD44^+^ cells appear in the periportal region, whereas MH-I-derived SHs appear in large numbers in the region adjacent to Thy1^+^/CD44^+^ cells. This indicates that MH-I-derived SHs (Thy1^−^/CD44^+^) are temporarily mixed in the periportal region adjacent to OC-derived SHs (Thy1^+^/CD44^+^) cells (Figure 3B). Therefore, the small bile ducts composed of OCs may regress when the number of hepatocytes is restored by both MH-II proliferation and MH-I-derived newly generated hepatocytes. Hence, liver regeneration is thought to be completed.

### 7.3. Exhaustion of Growth capability of MHs

With aging, MH-II reaches replicative senescence, and many MH-IIs become MH-IIIs. The number of MH-Is decreases accordingly. Furthermore, ActA expression is upregulated in the aging rat liver. This suppresses MH proliferation, so that MH-IIs cannot compensate adequately for the dying MH-IIIs by MH-II division, and MH-Is (SHs) are activated to form foci of SHs. Surgical liver removal of older animals, including humans, takes longer to recover to its original mass compared to younger ones. Conversely, during the cirrhotic stage of chronic liver disease in humans, the proliferative capacity of viable hepatocytes is known to be lost due to replicative senescence caused by telomere shortening. The replicative senescence of hepatocytes in patients with chronic liver disease caused by continuous hepatocyte division is due to continuous proliferative stimulation over 20–30 years. Thus, the entire regenerative nodules are composed of IHs in liver cirrhosis.

### 7.4. Massive Loss of MHs

Massive or submassive loss of MHs is typically observed in patients with ALF. The initial regenerative response is thought to be primarily mediated by surviving hepatocytes. IHs significantly increase in the injured liver if adequate adjuvant liver therapy, such as APOLT, is given when most hepatocytes are lost. This indicates that the differentiation process from DRs to MHs is occurring in the damaged liver.

## 8. Conclusions

As previously discussed, SHs emerge during liver regeneration when liver function is insufficient to maintain homeostasis. SHs continue to emerge until hepatocyte numbers and/or liver function are restored to a healthy state, but most are short-lived and many quickly differentiate into MHs. However, SHs are not a single population, but a mixture of cells of two different origins, MH-I-derived (MH-SHs) and OC-derived (OC-SHs). Both SHs share the basic characteristics of being smaller in size and less differentiated than typical MHs, but with some differences, as follows:

MH-SHs appear when relatively large numbers of MHs remain after liver injury.

Not only the remaining MHs but also OCs are activated when many MHs are injured, and both OC-SHs and MH-SHs transiently coexist in the lobule, but MH-SHs mainly contribute to the recovery of MH numbers. Most OC-SHs disappear by apoptosis.MH-SHs are relatively larger in size than OC-SHs, and MH-SHs are more differentiated than OC-SHs.OC-SHs are more differentiated into BECs than MHs, while MH-SHs are rarely differentiated into BECs.OCs are strongly activated, OC-SHs emerge when most MHs are damaged, and the majority of the liver lobules are occupied by OC-derived MHs. However, OC-derived MHs are gradually eliminated and eventually replaced by resident MH-derived cells as surviving MHs gradually proliferate via MH-SHs and regain sufficient function and original mass.MH-SHs transplanted into livers may be inserted into hepatic cords much more readily than OC-SHs.

We have studied “small hepatocytes” in vitro and in vivo, but many questions remain unanswered, such as what signals trigger the transition from MHs to SHs, what molecular mechanisms are involved, why CD44 is expressed in SHs, etc. Hence, further experiments are needed to elucidate the remaining issues.

## Figures and Tables

**Figure 1 cells-12-02718-f001:**
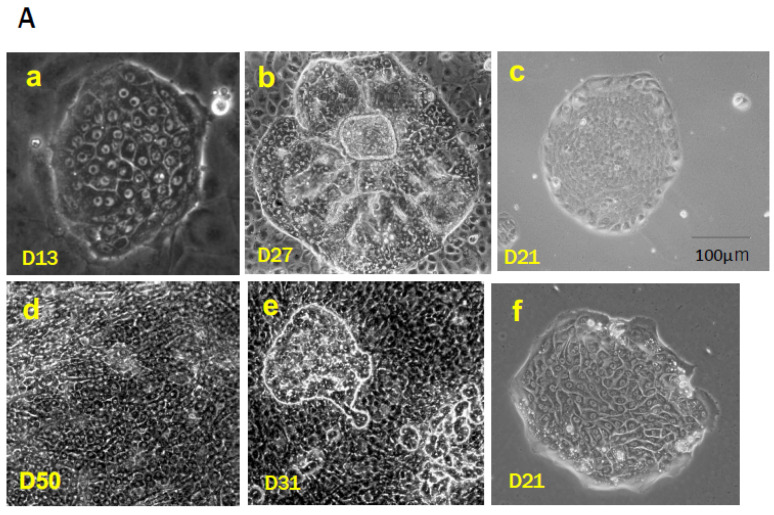
Small hepatocytes (SHs) in vitro and in vivo. (**A**) Photos of SHs isolated from a rat liver (**Aa**–**Ac**) and a human liver tissue (**Ad**–**Af**). A photo of an SH colony consisting of small-sized cells showing a flat surface 13 days after plating (**Aa**). A colony consisting of piled-up cells is surrounded by NPCs cultured for 27 days (**Ab**). A colony of rat HPPCs of the third passage cultured on thin-Matrigel 21 days after replating (**Ac**). Primary human hepatocytes isolated from a healthy part of the surgically dissected liver tissue are cultured for 50 days. Many mononuclear SHs are proliferating (**Ad**). Primary human hepatocytes proliferate to form a large colony and part of the cells pile up on the colony 31 days after plating (**Ae**). Primary human hepatocytes are cultured in the serum-free medium on a hyaluronan-coated dish for 21 days. A colony consisting of SHs is formed (**Af**). (**B**) Cells of an SH fraction isolated from a healthy rat liver are plated on the dishes coated with rat tail collagen (**Ba**), collagen type IV (**Bb**), fibronectin (**Bc**), thin-Matrigel (**Bd**), and collagen gel (**Be**,**Bf**). Cells were cultured in the modified Dulbecco’s modified Eagle medium (DMEM)/F12 medium with 10% fetal bovine serum (FBS) for 20 days. From 4 days after plating, 1% DMSO was added to the medium (**Ba**–**Bd**,**Bf**). Cells were fixed with absolute ethanol at day 20 and immunocytochemically stained with Krt8. Nuclei were stained with hematoxylin. SH colonies show Krt8-positive (Brown). (**C**) SHs appear in liver tissue (in vivo). (**Ca**) A cluster of small hepatocyte-like progenitor cells (SHPCs) is observed in the rat liver treated with retrorsine (Ret) and 2/3 PH (HE-staining). (**Cb**) A photo of the rat liver treated with d-galactosamine at 3 days after administration. (**Cc**) A focus of SHs observed in the 99-week-old rat liver. (**Cd**) Magnified image of the area surrounded by white lines in the photo (**Cb**). (**Ce**) A photo of the rat liver treated with d-galactosamine at 4 days after administration. (**Cb**,**Cd**,**Ce**) White arrowheads may indicate OC-SHs emerging near elongating bile ductules and yellow arrows may show MH-SHs near resident MHs. Bars show 50 μm.

**Figure 2 cells-12-02718-f002:**
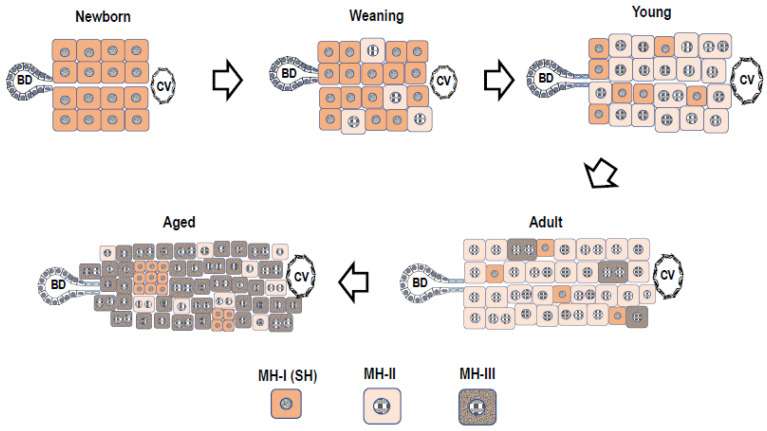
Subpopulations of MHs alter in their ratio and distribution with age. After birth, all hepatocytes are MH-I and actively proliferate to expand. At weaning, the heterogeneity of proliferative capability is manifested in hepatocytes, of which the proliferative capability is limited (MH-II). Most hepatocytes are MH-II and a small number of MH-I randomly remain in the lobules of an adult liver. With aging, MH-III emerge and gradually increase, which reach senescence, resulting in apoptosis. Some MH-I selectively proliferate to form foci in the aged liver.

**Figure 3 cells-12-02718-f003:**
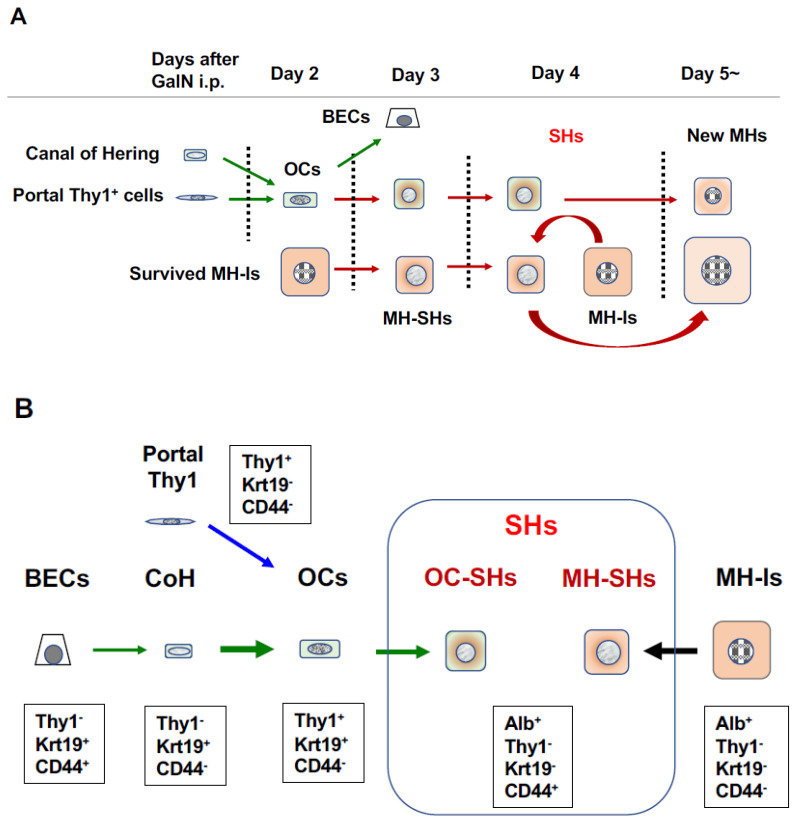
Origins of small hepatocytes in GalN-treated rat livers. (**A**) SHs are derived from OCs and MHs. GalN administration activates portal Thy1^+^ cells and the Canal of Hering to induce OCs, and then OCs differentiate into SHs and BECs within 3 days after the treatment. Conversely, the slightly damaged MHs are delayed to start their proliferation, and MH-SHs are derived from MHs. Some OC-SHs and the proliferated MH-SHs are differentiated into MHs. (**B**) Typical markers of the specific cell appeared in the process of liver regeneration by GalN administration.

**Figure 4 cells-12-02718-f004:**
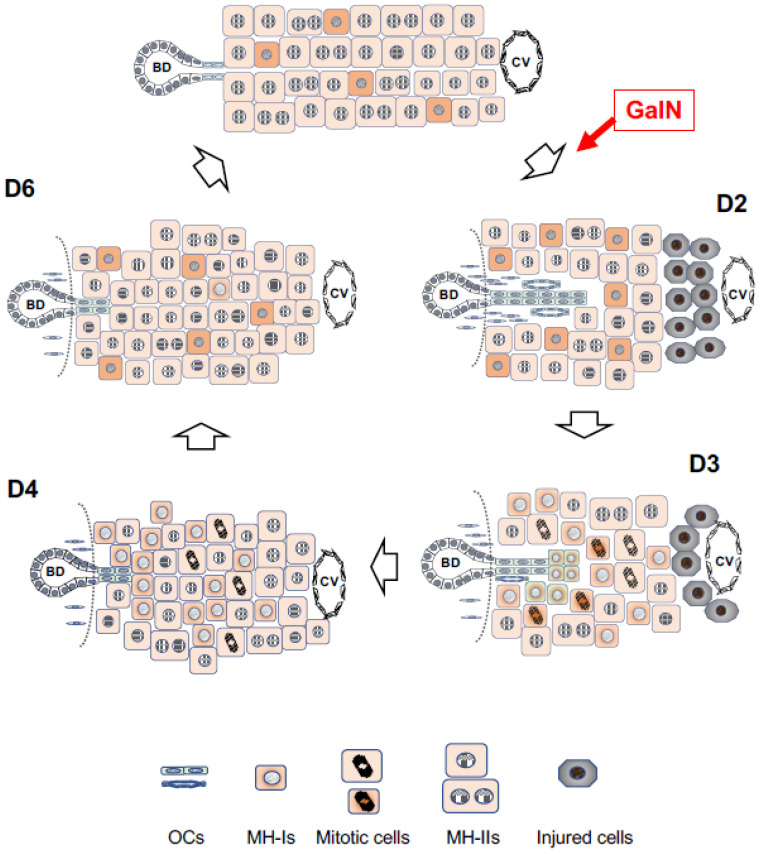
The appearance of small hepatocytes in the liver treated with GalN. OC-derived and MH-derived SHs sequentially appear at approximately 3 days after the treatment. MH-SHs majorly contribute to the reconstruction of the injured liver structure. A red arrow indicates that d-galactosamine (GalN) is intraperitoneally administered to a healthy rat. D2, D3, D4, and D6 show the day after GalN administration.

**Figure 5 cells-12-02718-f005:**
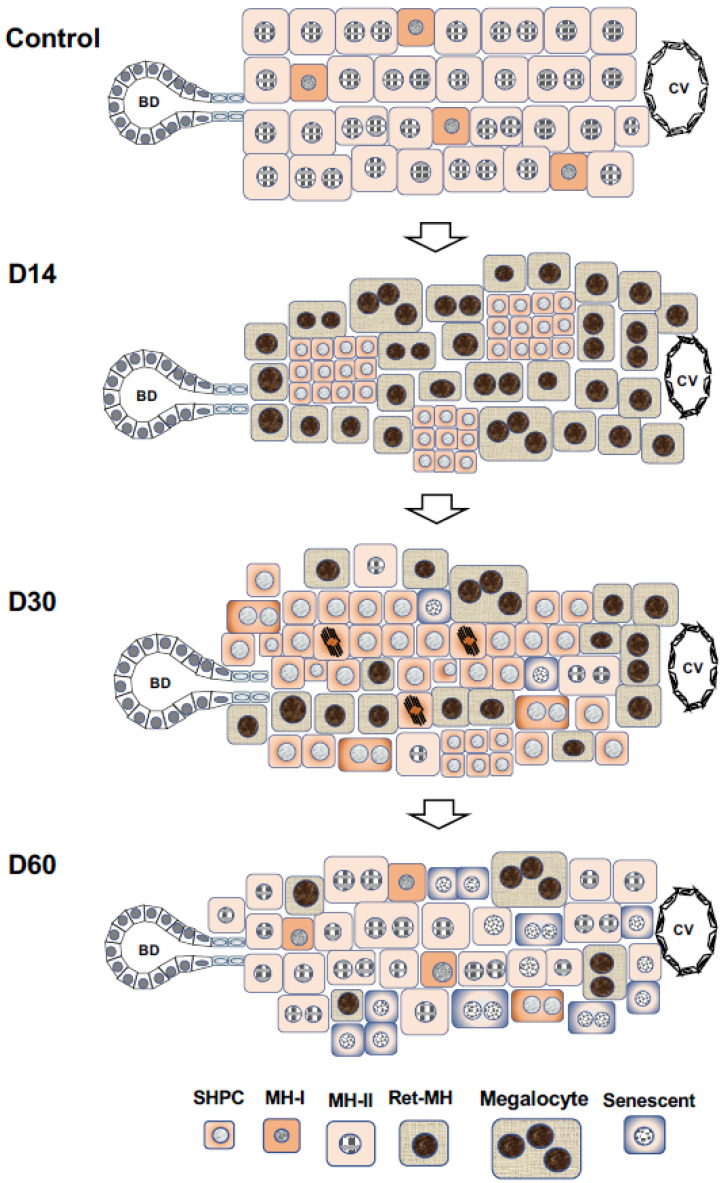
The emergence of small hepatocyte-like progenitor cells (SHPCs) in Ret/PH rat livers. Ret is intraperitoneally injected into rats at 2-week intervals, and 2/3 PH is performed 4 weeks after the second injection. SHPCs proliferate to form clusters, and SHPCs gradually differentiate into MHs 30 days later, and the liver mass is recovered. Thereafter, the number of replicative senescent hepatocytes increases. BD, bile duct; CV, central vein.

**Figure 6 cells-12-02718-f006:**
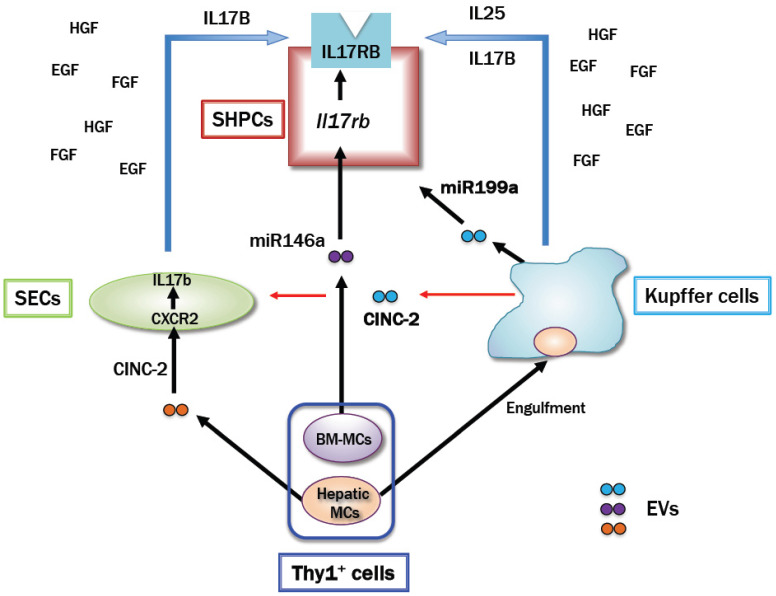
Growth regulation of SHPCs in Ret/PH rat livers by Thy1-positive cell transplantation. Thy1^+^ cells isolated from GalN-treated rat livers (Hepatic MCs) or BM-derived mesenchymal cells (BM-MCs) were transplanted via the spleen. SHPC proliferation was stimulated by cytokines and extracellular vesicles (EVs) produced by Thy1^+^ cells. Growth factors, such as EGF, HGF, and FGFs, are present in the liver after PH.

**Table 1 cells-12-02718-t001:** Characterization of the epithelial cells that emerged in the livers.

	Hepatoblasts	BECs	Oval Cells (Rodent)	HPPLs	IHBCs (Human)			SHs (MH-I)		MHs (MH-I/II/III)
					DRs	IHs (Ductular Hepatocytes)	HPPCs	OC-Derived	MH-I-Derived	
Ploidy	2n	2n	2n	2n	2n	2n	2n	2n	2n	>2n
No. of Nuclei	Mono	Mono	Mono	Mono	Mono	Mono	Mono	Mono	Mono	Mono/Binuclei
AFP	++	−	+	ND	−	−/+	+	−	−	−
DLK1	++	−	+	−	−	+	+	−	−	−
EpCAM	+	++	+	−	+	+/−	±	−	+	−
NCAM	−	−	+	ND	+	−	+		−	−
Krt7	+	+	+	−	−	+	±	−	−	−
Krt19	+	++	++	+	++	−	±	−	−	−
OV6	−	++	++	−	+	+	ND	−	−	−
Sox9	−	++	−	ND	UK	UK	±	−	−	−
Thy1	+/−	−	+	ND	UK	UK	+	+	−	−
CD44	+	++	−	ND	UK	UK	++	++	++	−
Integrin beta1	+	++	+	+	UK	+	++	++	++	−
HNF4α	+	−	−	+	+	+	+	+	++	++
Krt8/18	+	+	−	+	+	+	+	+	+	++
Albumin	+	−	−	+	+	+	++	++	++	++
References	[105,107,119,125]	[15,126,127,128,129,130]		[106]	[22,125,128,131,132,133,134]		[59,100]	[89,91,92]		
ND: Not Determined		UK: unknown								
BECs, biliary epithelial cells;										
HPPLs, hepatic progenitor cells proliferating on laminin;										
IHs, intermediate hepatocytes;										
IHBCs, intermediate hepatobiliary cells										
HPPCs, hepatocytic parental progenitor cells;										
SHs, small hepatocytes; MHs, mature hepatocytes										

**Table 2 cells-12-02718-t002:** Growth regulators of small hepatocytes.

				Factors	References
Paracrine	Stimulators	Mitogens	Growth factors	Hepatocyte growth factor (HGF)	[9,48,191]
				Epidermal growth factor (EGF)	
				Transforming growth factor (TGF)-a	
				Acidic fibroblast growth factor (aFGF/FGF1)	
				Basic fibroblast growth factor (bFGF/FGF2)	
				Pleiotrophin	[192]
		Co-mitogens	Cytokines	IL-17B	[158]
				IL-25	[158]
				IL-6	[171,193]
				Stem cell factor (SCF)	[171]
			microRNAs (miRs)	miR-146a-5p	[171]
				miR-125b-5p	[194]
				miR-199a-5p	[194]
	Inhibitors		Growth factors	Activin A	[195,196]
				TGF-β	[195]
			Cytokines	Interleukin 1β (IL-1β)	[195]
			Others	* Dexamethasone	[193]
Autocrine	Stimulators		Growth factors	TGF-α	[191]
				Follistatin	[196]
	Inhibitor			Activin B	[196]

* SHPCs.

## Data Availability

Not applicable.

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
