# Peer review of "“Small Hepatocytes” in the Liver"

_cells, 2023, doi:10.3390/cells12232718_

Round 1

Reviewer 1 Report

Comments and Suggestions for Authors

In this review, the authors devlop the concept of the role of small hepatocytes in liver regeneration. Briefly, they differentiate between type I cells (MH-I), which are progenitor cells with proliferative capacity; type II with a limited proliferative capability; and type III (MH-III) cells that lost the ability to divide.

The review contributes an interesting discussion on the role of hepatocytes in liver regeneration. However, the authors should address the following alternative concept and describe the experiments that verify their (above described) concept. An alternative concept (tot he „small hepatocyte concept“) is that all hepatocytes in the liver lobule – independent of their size – in principle have the capacity to proliferate and by this may contribute to liver regeneration. In principle, this concept can easily be tested experimentally. One of many possibilities to track proliferating hepatocytes is to induce liver damage and administer BrdU that incorporates into the DNA of proliferating hepatocytes. After a single intoxication with a hepatotoxic dose (e.g. 1 g/kg) CCl4, more than 30% of the hepatocytes (distrubuted all over the liver lobule) become BrdU positive: https://doi.org/10.1073/pnas.0909374107.

Upon repeated CCl4 intoxication (and BrdU administration) some hepatocytes should remain BrdU negative, if the „small hepatocyte concept“ is correct; in contrast, all hepatocytes should become BrdU positive after repeated intoxication/regeneration cycles, if all hepatocytes have the capacity to proliferate. It would be appreciated, if the authors would introduce their concept by describing the easiest and most convincing experiments that led them to their conclusions.

Minor: the authors should perform a careful language check to remove minor errors; an example in the abstract is: „type II cells (MH-II), which possess a limited proliferative capability, and type III cells (MH-III), which loses the ability to divide..“. Should be: „…which lose the ability…“

Comments on the Quality of English Language

Minor: the authors should perform a careful language check to remove minor errors; an example in the abstract is: „type II cells (MH-II), which possess a limited proliferative capability, and type III cells (MH-III), which loses the ability to divide..“. Should be: „…which lose the ability…“

Author Response

Dear Reviewer 1,

Thank you very much for your valuable comments.

We have revised the manuscript according to the Reviewers’ as well as the Editor’s comments and suggestions.

Major changes in this revised manuscript include in the following:

  1. Added Graphical Abstract and modified Figures 3, 4, and 6.
  2. Added a new table (Table 1).
  3. Added references in the appropriate sites of the revised manuscript, including the papers that were required by Reviewer 2. Therefore, the number of references increased from 187 to 214.

Please see our point-by-point responses to your comments below.

In this review, the authors develop the concept of the role of small hepatocytes in liver regeneration. Briefly, they differentiate between type I cells (MH-I), which are progenitor cells with proliferative capacity; type II with a limited proliferative capability; and type III (MH-III) cells that lost the ability to divide.

The review contributes an interesting discussion on the role of hepatocytes in liver regeneration. However, the authors should address the following alternative concept and describe the experiments that verify their (above described) concept. An alternative concept (to the „small hepatocyte concept“) is that all hepatocytes in the liver lobule – independent of their size – in principle have the capacity to proliferate and by this may contribute to liver regeneration. In principle, this concept can easily be tested experimentally. One of many possibilities to track proliferating hepatocytes is to induce liver damage and administer BrdU that incorporates into the DNA of proliferating hepatocytes. After a single intoxication with a hepatotoxic dose (e.g. 1 g/kg) CCl4, more than 30% of the hepatocytes (distributed all over the liver lobule) become BrdU positive: https://doi.org/10.1073/pnas.0909374107.

Upon repeated CCl4 intoxication (and BrdU administration) some hepatocytes should remain BrdU negative, if the „small hepatocyte concept“ is correct; in contrast, all hepatocytes should become BrdU positive after repeated intoxication/regeneration cycles, if all hepatocytes have the capacity to proliferate. It would be appreciated, if the authors would introduce their concept by describing the easiest and most convincing experiments that led them to their conclusions.

We greatly appreciate your valuable suggestions. Please see our thoughts on your suggestion below.

We have not performed the experiments indicated by the reviewer, thus we cannot verify whether the unlabeled cells that resulted from frequent CCl4 administration are label-retaining cells, cells that reach to replicative senescent, or cells that are affected by the drug. Currently, it is difficult to conclude that the presence of unlabeled cells that were shown by the experiments proves our hypothesis.

We categorized hepatocytes according to their proliferative capacity into three types: MH-I, MH-II, and MH-III. Almost all hepatocytes, which are isolated from two- and three-week-old rat livers, multiply or divide to proliferate in a culture until six days after plating; hence, we could not recognize the difference in growth capacity among hepatocytes. However, we could distinguish the cells, among hepatocytes that were isolated from ≥4-week-old rat livers that have a higher proliferative capability to form colonies consisting of small hepatocytes (SHs) from other hepatocytes. 

The frequency of SHs to form colonies rapidly decreased with age, reaching < 2% at 8-15 weeks of age. The frequency of SHs in rat hepatocytes around 100 weeks decreased to approximately 0.5%, and the number of cells that were not labeled at all increased, as well as cells that died during culture. Further, we have confirmed, although not quantified, that the number of SA-b-Gal-positive cells increases with increasing age in weeks in adult rat livers. Therefore, we hypothesize that hepatocytes can be categorized into three types by their growth capabilities after birth. MH-I, in an emergent situation for animals, becomes SHs as progenitor cells, thereby rapidly generating new hepatocytes. Approximately 20% of SHs act as progenitor cells with self-renewability, which we name HPPCs.

SHs transiently appear in various liver pathological conditions. Further, SHs were considered as a temporary form of differentiation process from liver stem cells, OCs (DRs), to hepatocytes, or as an intermediate form of metaplasia from hepatocytes to biliary epithelial cells (BECs). Our experimental data have indicated that SHs are cells that emerge from MHs to increase their proliferative capacity by temporarily suppressing their differentiation functions in order to rapidly increase their cell number, and then revert to MHs when their role is completed. Conversely, we have revealed that special conditions are required for the conversion of SHs into BECs. Additionally, the differentiation from OCs to MHs occurs under conditions in which resident hepatocytes have difficulty proliferating and maintaining homeostasis. OCs-derived hepatocytes may be eliminated once their resident hepatocytes can proliferate, thereby recovering their liver functions. We concluded that this review considering liver regeneration from the point view of SHs based on our results and those reported by other researchers.

Minor:

The authors should perform a careful language check to remove minor errors; an example in the abstract is: „type II cells (MH-II), which possess a limited proliferative capability, and type III cells (MH-III), which loses the ability to divide. “. Should be: „... which lose the ability...“

Thank you very much for your meticulous review.

In the revised manuscript, we have corrected the points that you have emphasized.

Reviewer 2 Report

Comments and Suggestions for Authors

This is an informative and interesting review based on the authors’ long-term research. This review will help us understand the small hepatocytes and its role in liver regeneration. Before acceptation, the following questions are needed to be clarified by the authors.

1. P7 Line 252-258, Figure2, Have the authors compared the differences in SHs functions (such as proliferation and differentiation) between young mice and aged mice? Did the SHs of aged mice exist in the liver from newborn and continue to divide to adult? or was it later derived from MH?

2. P14 Figure 3B, the author proposes that there are three types of MH: MH-I, MH-II and MH-III. MH-I is SH. In the figure, MH becomes MH-SH, but does MH-II or MH-III become MH-SH? Please describe it clearly. How did SH originate?

3. P16 Figure 4, please mark what each kind of cell is in the figure. Figure 2 and Figure 5 are clearly marked.

4. P9 Line 376-377, Are there any other distinguishing molecular markers between SH and MH besides CD44? For example, how different is the expression level of Ki67? Is Human SH also CD44 positive cells?

5. P10 Line 427-428, P11 Line P12. Hepatocyte parental progenitor cells (HPPCs) is SHs? Could the authors arrange a table to compare the characteristics (morphology, mononuclear or binuclear or polynucleated, hepatocyte function, etc.), the differences and the cell markers (such as CD44, DLK, CK19, Thy, etc.) of SH, HPPCs, hepatoblasts, HSPCs (hepatic stem/progenitor cells), HPPLs and IHs (intermediate hepatocytes)? It will help the reader have a clear concept to these cells.

Minor concern:

1. P3 Line117-118, 'Oval cells (OCs) and 117 small hepatocytes (SHs) are respectively known as liver stem and progenitor cells.' Please add the references.

2. P3 Line 119, 'OCs express markers of biliary epithelial cells (BECs), hepatoblasts, and hematopoietic stem cells (19,44,60,84,127). " Could the authors details what markers of BECs hepatoblasts and and hematopoietic stem cells?

3. P3 Line 129, Please change "- fetoprotein" into "α-fetoprotein".

4. P4 Line180, here "27" means "27 colonies"?

5. P6 Line236, "28 cell-doublings" means "28 cell-population-doubling"?

6. P6 Line242-243, are MH-III cells senescent hepatocytes?

7. P12 Line501-503, Please provide the references.

8. P12 Line507-508, Please provide the references.

9. P12 Line508-509 Recent articles have shown that hepatocytes with no specific diploid / high expression of a gene have higher mitotic ability, and the homeostasis of the liver and the repair after injury require the joint action of many hepatocytes. What does the authors think of this view?

Refs: 1. 2020. Broad Distribution of Hepatocyte Proliferation in Liver Homeostasis and Regeneration. 

2. 2019. In Vivo Lineage Tracing of Polyploid Hepatocytes Reveals Extensive Proliferation during Liver Regeneration.

3. 2019. AXIN2+ Pericentral Hepatocytes Have Limited Contributions to Liver

Homeostasis and Regeneration.

10. P14 Line577-582 Krt19 is CK-19 the author needs to use one form not two forms.  

11. P18 Line664 Please revise Figure 5. 'he' into' The'

Author Response

(The authors gave the same response as above.)

Round 2

Reviewer 1 Report

Comments and Suggestions for Authors

The authors have carefully revised the manuscript. From my point of view it is acceptable for publication.